# COMPOSITIONAL TASK REPRESENTATIONS FOR LARGE LANGUAGE MODELS

**Nan Shao**[*1], **Zefan Cai**[*2], **Hanwei Xu**[1], **Chonghua Liao**[3], **Yanan Zheng**[3], **Zhilin Yang**[†3451]

[1]Recurrent AI, [2]Beijing Jiaotong University, [3]Tsinghua University
[4]Shanghai Artificial Intelligence Laboratory, [5]Shanghai Qi Zhi Institute
{windoker,zefncai}@gmail.com
{zyanan, zhiliny}@tsinghua.edu.cn

## ABSTRACT

Large language models have shown a remarkable cross-task generalization ability. Most prior works assumed that prompts effectively extract knowledge from language models to facilitate generalization to new tasks. This perspective led to numerous studies on improving prompts. In contrast, we introduce a new perspective, compositional generalization, that views each task as a composition of latent codes and generalizes to test tasks by a new composition of seen codes. To this end, we propose a novel prompt-free approach, Compositional Task Representations (CTR), that employs multi-task training to learn a discrete, compositional codebook. Empirically, our CTR substantially outperforms prompt-based methods in zero-label learning on average. According to our analysis, some of the learned CTR codes are interpretable to humans and demonstrate a certain degree of controllability.

## 1 INTRODUCTION

Large language models (LLMs) have shown remarkable performance in cross-task generalization. Without using any labeled data for the target task, GPT-3 (Brown et al., 2020) obtains reasonable performance on a wide range of tasks. Later extensions such as FLAN (Wei et al., 2022) and T0 (Sanh et al., 2022) continue training the LLMs on a large number of supervised tasks, which further improves cross-task generalization performance. The aforementioned studies have used an important assumption that natural language prompts extract knowledge from LLMs to facilitate generalization to new tasks. In this direction, numerous studies have focused on different aspects of improving prompt-based learning, such as designing better prompts (Xu et al., 2022), increasing the number of prompts (Wang et al., 2022; Aribandi et al., 2022), and improving the training efficiency of prompts (Lester et al., 2021).

In contrast, we explore an alternative perspective for cross-task generalization, i.e., compositional generalization. Specifically, we explore whether it is possible to represent tasks using discrete compositions of latent codes. This perspective enjoys several potential benefits. First, since the latent codes have been trained for seen tasks, we expect the LLMs to have strong cross-task generalization abilities because new tasks can also be represented as a composition of these trained codes. Second, it provides a way to analyze and understand cross-task generalization by investigating the association between tasks and the learned representations. Third, it has the potential of being more controllable than prompts for task generalization due to its built-in compositionality.

Motivated by the aforementioned potentials, we propose a new method, Compositional Task Representations (CTR), that employs multi-task training to learn a discrete, compositional codebook. Specifically, given a large number of training tasks, we use an encoder to map each randomly-initialized task embedding to a fixed-length sequence of query vectors. Each query vector is used to retrieve a code from a codebook, which is formulated as an embedding lookup table. This produces

---

[*]Equal contribution.
[†]Corresponding author.
[‡]The code will be available at https://github.com/shaonan1993/CTR.

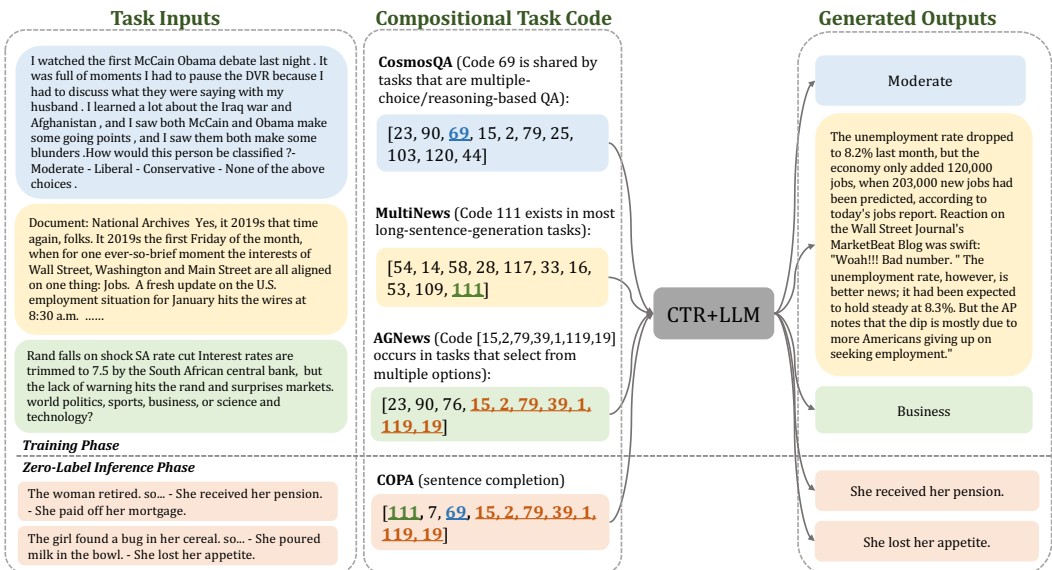

Figure 1: An illustration of how CTR generalizes to zero-label tasks. In this real example produced by our model, CTR combines the abilities of reasoning-based QA, sentence generation, and multi-choice selection from training tasks to perform a new task COPA.

a sequence of codes, a compositional representation of the current task. These compositional codes are fed as the input to an LLM in place of prompts to make predictions. At test time, given a new task, we use unlabeled data to search for a high-performing composition of codes, which enables zero-label cross-task generalization. CTR is also applicable to the few-shot setting where the few labeled examples are used for code search.

Empirically, we demonstrate improved performance under both the settings of zero-label learning and few-shot learning, outperforming strong baselines including prompt tuning, model tuning, and genetic prompt search (Xu et al., 2022). Importantly, we analyze the learned task representations and show that they demonstrate a certain degree of interpretability and controllability. For example, as shown in Figure 1, CTR learns to generalize to a new task by a new composition of existing codes.

## 2    RELATED WORK

**Language Model Prompting.**    Brown et al. (2020) showed that GPT-3 performs well in the few-shot setting if properly handcrafted prompts are provided. Other works (Shoeybi et al., 2019; Rae et al., 2021; Schick & Schütze, 2021) also report promising zero-shot or few-shot performances of LLMs. Besides, Wei et al. (2022) and Sanh et al. (2022) collect a set of labeled datasets and use manual templates to transform them into a sequence-to-sequence style. Such a formulation makes it possible to continue training LLMs on these labeled datasets and improves cross-task generalization. Wang et al. (2022); Mishra et al. (2022) introduced a benchmark of over 1600 tasks and their expert-written instructions Gao et al. (2021); Shin et al. (2020) studied automating the search process of discrete prompts. Li & Liang (2021); Liu et al. (2021) propose continuous soft prompts with gradient-based optimization. Compared to these approaches, we study a different approach that learns compositional task representations, which benefits cross-task generalization.

**Compositional Architecture for LLMs.**    Previous work has explored designing compositional architectures. Sparsely Gated Mixture of Expert (MoE) (Lepikhin et al., 2021) activates a subset of a network given the input data. Artetxe et al. (2021) trained an MoE model with 1.1T parameters, which is shown to outperform a dense model with similar computational cost. SkillNet-NLU (Tang et al., 2022) and SkillNet-NLG (Liao et al., 2022b) employed a similar sparsely activated mechanism to handle different NLU or NLG tasks. Different from these approaches, our approach focuses on learning compositional task representations using a discrete codebook.

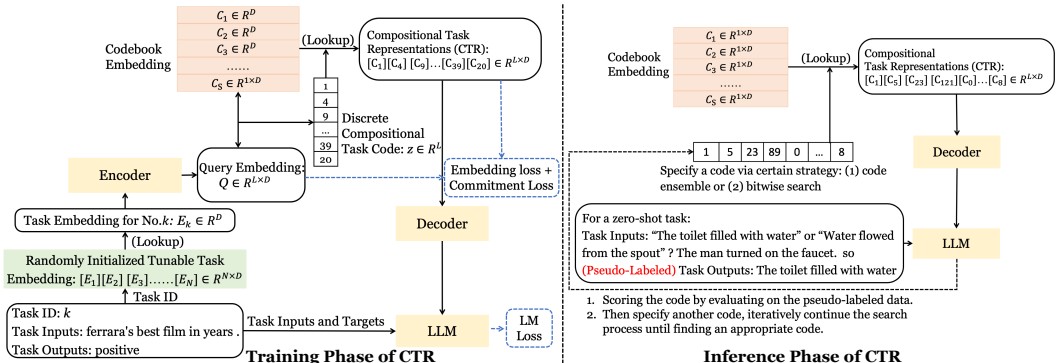

Figure 2: An overview of the architecture of our proposed CTR . The left part illustrates the training phase, while the right part shows how CTR works during the inference phrase.

## 3 COMPOSITIONAL TASK REPRESENTATIONS

The motivation of CTR is to explore the cross-task generalization ability of LLMs from a brand-new perspective—compositional generalization, and to further improve the performance of cross-task generalization. Specifically, our main hypothesis is that being trained on a variety of natural language tasks, LLMs will be able to learn to represent each task as a composition of discrete latent codes, where each latent code is associated with certain aspects of a task. As a result, CTR potentially enjoys the advantages of better cross-task generalization since it could represent new tasks by forming new compositions.

This section introduces the overall architecture of CTR and how it is trained to overcome optimization challenges. As Figure 2 shows, CTR consists of the CTR learning module and an LLM, where the CTR learning module contains an encoder, a decoder, task embeddings, and a codebook.

### 3.1 DISCRETE LATENT TASK CODEBOOK

We define a latent task codebook embedding space $C \in R^{S \times D}$ where $S$ is the size of the latent codebook embedding space (i.e., each task code can take either of the $S$ categorical values), and $D$ is the dimension of each latent code embedding $C_i \in R^D, i \in [1, 2, \cdots, S]$. This is analogous to the idea of VQ-VAE (van den Oord et al., 2017) where a discrete latent codebook is also employed.

As Figure 2 shows, given a training task as the input (assuming the task id is $k$), CTR first obtains its task embedding $E_k \in R^D$ by retrieving from a randomly-initialized task embedding lookup table $R^{N \times D}$ where $N$ is the number of training tasks. The task embedding $E_k$ is then passed to the encoder module and is mapped into a fixed-length sequence of query vectors $Q \in R^{L \times D}$ where $L$ is the length of the sequence. For each query vector $Q_l, l \in [1, 2, \cdots, L]$, it is used to retrieve a task code embedding from the codebook embedding space. Specifically, it calculates the $l_2$ distance with each of the latent code embedding $C_i \in R^D, i \in [1, 2, \cdots, S]$, and find the nearest neighbor,

$$\text{CTR}_l = C_{z_l}, \quad \text{where } z_l = \arg\min_i ||Q_l - C_i||_2 \tag{1}$$

In this way, all query vectors together produce a $L$-length sequence of code embeddings, denoted as the compositional task representations (CTR). They are then passed through a decoder, followed by being used as the input to an LLM in place of prompts to make predictions.

Let $\mathbf{z}$ be a vector of $L$ latent codes $z_1, z_2, \ldots, z_L$. We consider each latent code as describing an attribute or a necessary skill of a certain task, such as task type, output space, etc. Intuitively, we need multiple codes to fully describe a task, and each task is formulated as a composition of codes.

### 3.2 TRAINING

Training can be challenging due to the existence of discrete latent variables. Moreover, during the initial training phase, the CTR learning module is randomly initialized. As a result, the codebook embeddings $C$ will have a very different distribution to the query vectors $Q$, which increases the

difficulty of optimization. We decouple training into two phases. In the first phase, we freeze the LLM and only update the CTR learning module. This is followed by tuning all parameters.

In terms of the loss function, following van den Oord et al. (2017), we employ two separate losses, an embedding loss and a commitment loss, to match the query vectors with the compositional task representations. These losses are used in combination with a standard language modeling loss,

$$L = L_{\text{LM}} + \sum_{l=1}^{L} (||\text{sg}[Q_l] - \text{CTR}_l||_2^2 + \beta ||Q_l - \text{sg}[\text{CTR}_l]||_2^2) \tag{2}$$

where $L_{\text{LM}}$ is a standard language modeling loss for solving the target task, sg denotes stopping the gradients, and $\beta$ is a hyperparameter.

### 3.3 INFERENCE

We consider two settings: zero-label learning and few-shot learning, and describe how we apply CTR in these settings.

**Code Ensemble for Zero-Label Learning.** In the zero-label setting, we are given a new task along with a set of unlabeled data. * The question is how to decide the code for this new task without labeled data. Our main idea is to select one of the training tasks (319 in total in our experiments) and use its learned code for this new task. We first obtain a set of candidate codes by examining how much a code gives predictions that deviate from a uniform distribution on the unlabeled data (Zhao et al., 2021). The candidate set is formed by $\mathcal{N}$ (set to 60 in our experiments) codes with the lowest deviations. We then ensemble the candidate set of codes to predict pseudo labels on unlabeled data, and select the code with the highest pseudo-label accuracy.

**Bitwise Search for Few-Shot Learning.** In the few-shot setting, we are given a new task along with a set of labeled data. We use this set of labeled data as our validation set to search for a high-performing code. Our preliminary study shows that one can control the output of CTR by changing one bit in the code vector $\mathbf{z}$. Inspired by this, we first examine the validation-set accuracy for codes of training tasks, and select the code with the best accuracy as an initialization. Then we iteratively change a single bit of the selected code and evaluate the validation-set performance. At each iteration, we keep the updated code with higher performance. Finally, the code vector that obtains the best result on the validation set is taken as the test-task code. Our preliminary study shows that a certain code value usually occurs in a specific position of the code vector, and each position usually only has a small set of code values. Motivated by this, we only changes a small set of valid code candidates for each position.

In both the zero-label and few-shot settings, after we obtain a code vector $\mathbf{z}$ for the new task, we use the vector to obtain a composition task representation by indexing $\mathbf{z}$ in the codebook $C$. The task representations are then passed through the decoder and the LLM to perform the new task, similar to training time. This is also illustrated in the right part of Figure 2.

## 4 EXPERIMENTS

### 4.1 EXPERIMENTAL SETUP

We conduct extensive quantitative experiments to validate the performance of cross-task generalization of CTR. We mainly consider two settings, the zero-label setting and the few-shot setting. Aside from quantitative experiments, we perform qualitative analysis to understand the cross-task generalization ability by investigating the association between the discrete latent codes and the tasks.

### 4.1.1 DATASETS

CTR requires a large multi-task set for training, and a held-out set of tasks whose types are never seen during training for evaluation. We follow the T0 benchmark. The training part consists of

---

*To clarify, for the zero-label setting, the unlabeled data equals the inputs of the test task. Note that the inputs naturally exist as long as we perform the evaluation. So it requires no extra effort to acquire data.

39 tasks of 8 task types, including closed-book question answering (QA), multiple-choice QA, extractive QA, sentiment analysis, paraphrase identification, topic classification, summarization, and structure to text. The test part consists of 11 tasks of 4 task types, including natural language inference (RTE (Dagan et al., 2006), CB (De Marneffe et al., 2019), ANLI/R1-R3 (Nie et al., 2020)), coreference resolution (WSC (Levesque et al., 2012), Winogrande (Sakaguchi et al., 2020)), sentence completion (COPA (Roemmele et al., 2011), StoryCloze (Mostafazadeh et al., 2017), Hellaswag (Zellers et al., 2019)), and word sense disambiguation (WiC (Pilehvar & Camacho-Collados, 2019)). Both the training and test parts are disjoint in task types, ensuring the zero-label setting. We follow T0 to use the accuracy on the validation split of test tasks as the metric.

### 4.1.2 BASELINES

For the zero-label setting, we primarily compare CTR with the following baselines. It is noteworthy that all baselines share a similar model size as CTR (i.e., about 770M), thus are comparable. We provide implementation details of baselines in Appendix A.7.

• **T0** (Sanh et al., 2022) shares similar goals as CTR, which uses prompted multi-task training to improve the generalization performance. T0 reports the average results over multiple prompts.[†].

• **Self-Training** Since unlabeled data are accessed, we consider the self-training (Schick & Schütze, 2020) method as one of the baselines. Starting from T0, it uses T0 to label the unlabeled data, and further finetunes T0 with the pseudo-labeled data. It also reports average performance over prompts.

• **Manual-Code** uses artificial feature vectors in place of CTR vector. Specifically, we manually label a set of artificially-designed features for each task, including the number of input fields of the task, whether it requires reasoning, and whether it is a classification task, etc. Each task can be represented as an discrete feature vector, each dimension associated with one of the aspects.

• **ZPS** (Liao et al., 2022a) is method for zero-label prompt selection. It first labels a set of unlabeled data through prompt ensemble, and use the pseudo-labeled data to select the best natural language prompt for the test task. We apply ZPS to multi-task T0 as a baseline.

For the few-shot setting where there are 32 test-set labeled examples, based on the multi-task T0, we experimented the following five baseline methods.

• **Model Tuning** directly finetunes the pretrained model using the test-set labeled data. Specifically, we follow the few-shot setting in Zheng et al. (2021) to use 16 labeled data for finetuning and another 16 labeled samples for model selection.

• **Prompt Tuning** (Lester et al., 2021) introduces additional continuous prompts to the backbone model (i.e., T0) and trains the continuous prompts using the few labeled data.

• **GPS** (Xu et al., 2022) is a genetic prompt search method. Based on T0, GPS gradually mutates the prompts with a generative model and uses the few labeled data to selects prompt candidates.

• **GRIPS** (Prasad et al., 2022) is a gradient-free edit-based method for optimal prompt search.

• **Black-Box Tuning (BBT)** (Sun et al., 2022) is a gradient-free few-shot prompt selection method. Unlike GPS and GRIPS, it searches for the best soft prompt embedding in a continuous space.

### 4.1.3 TRAINING DETAILS

We instantiate our CTR with T5-Large (Raffel et al., 2019) being the LLM. We implement both the encoder and the decoder in our CTR model as two linear networks. We have experimented with various architectures (see Appendix A.4 for more details). For the first stage of training where the LLM is frozen, we use the Adam optimizer with a learning rate of 1e-2, a decay rate of 0.1, and a batch size of 2048. We use a codebook embedding dimension of 1024, which is the same as the hidden dimension. The CTR length is set at 10 and each position can be assigned values ranging from 0 to 127; i.e., the codebook size is 128. The hyperparameter $\beta$ is set at 0.1. We experimented with different codebook sizes and CTR lengths, and detailed results are provided in Appendix A.5. For the second training phase where all parameters are updated, we use the Adam optimizer with a learning rate of 1e-4 and a batch size of 1024. We follow the training recipe as T0 (Sanh et al., 2022) for the rest of the hyperparameters.

---

[†]T0 uses natural language prompts from PromptSource (Bach et al., 2022).

| Method | Natural Language Inference | | | | | Sentence Completion | | | Co-reference | | WSD | Avg. |
|---|---|---|---|---|---|---|---|---|---|---|---|---|
| | RTE | CB | ANLI1 | ANLI2 | ANLI3 | COPA | Hella. | Story. | WSC | Wino. | WiC | |
| Zero-Label Setting (unlabeled data of each test task) | | | | | | | | | | | | |
| T0-Large | 72.67 | 56.55 | 32.77 | 32.15 | 34.38 | 85.36 | 27.18 | 93.04 | 63.94 | 54.35 | 50.33 | 54.79 |
| Self-Training | 73.57 | 76.14 | 34.42 | 32.90 | 37.44 | 87.45 | 30.33 | 94.54 | 57.08 | 56.56 | 50.75 | 57.38 |
| Manual-Code | 75.19 | 56.89 | 33.12 | 32.49 | 33.48 | 75.76 | 30.84 | 93.10 | 61.16 | 54.10 | 51.45 | 54.33 |
| ZPS | 79.06 | 67.86 | 31.20 | 31.10 | 34.25 | 88.00 | 29.16 | 93.43 | 65.38 | 53.43 | 49.84 | 56.61 |
| Our CTR | 80.51 | 87.50 | 33.40 | 34.40 | 33.80 | 92.00 | 27.50 | 90.10 | 56.58 | 49.40 | 62.50 | **58.88** |
| Few-Shot Setting (32 labeled data of each test task) | | | | | | | | | | | | |
| Model Tuning | 75.31 | 80.95 | 35.73 | 31.31 | 35.93 | 82.05 | 41.86 | 92.04 | 55.96 | 56.74 | 52.15 | 58.18 |
| Prompt Tuning | 77.08 | 76.90 | 31.89 | 31.86 | 35.53 | 81.70 | 31.18 | 94.10 | 62.88 | 55.42 | 51.22 | 57.25 |
| GPS | 77.68 | 79.64 | 32.71 | 31.49 | 37.56 | 81.08 | 28.11 | 93.40 | 64.23 | 52.72 | 52.52 | 57.38 |
| GRIPS | 71.56 | 70.89 | 32.14 | 32.26 | 34.77 | 77.56 | 26.44 | 93.40 | 62.12 | 52.96 | 52.12 | 55.11 |
| BBT | 71.19 | 57.26 | 33.79 | 32.00 | 35.30 | 76.49 | 28.95 | 93.11 | 62.12 | 53.40 | 52.93 | 54.23 |
| Our CTR | 80.51 | 83.93 | 34.40 | 34.20 | 36.60 | 89.00 | 35.07 | 91.70 | 68.18 | 55.00 | 58.62 | **60.66** |

Table 1: Main results of CTR and baselines on 11 test tasks under the zero-label setting and the few-shot setting. The zero-label setting allows using unlabeled data of the test task while the few-shot setting uses 32 labeled data of the test task. All methods share a similar model size (i.e., 770M).

## 4.2 MAIN RESULTS AND ANALYSIS

The performance of cross-task generalization, respectively under the zero-label setting and the few-shot setting, are shown in Table 1. Our CTR outperforms all baseline methods on average under both settings. Comparing CTR with T0-Large and its variant (i.e., self-training), CTR outperforms them respectively by more than 4 points and by almost 1.5 points on average. The improvements potentially originate from two aspects—(a) the learned compositional task representations benefit from better generalization abilities than the discrete manual prompts used by T0/self-training; (b) our CTR can select a high-performing compositional task representation for the unseen task. Compared with Manual-Code, our CTR demonstrate significant advantages of more than 4.5 points, proving that artificially-designed features of tasks are unreliable, and CTR provides an effective way of automatically training such data-driven compositional task representations using multi-tasks.

Compared with Model Tuning, Prompt Tuning, and BBT, where they require parameter updates over the test-task data, CTR shows better performance of cross-task generalization on average. Compared with baselines that do not tune parameters (i.e., GPS, GRIPS), CTR shows even larger and more consistent advantages on most of the test tasks; i.e., on 9/11 tasks CTR shows dominating performance.

On co-reference tasks , CTR performs worse under the zero-label setting and better under the few-shot setting. The zero-label setting uses pseudo-labeled data to select test codes while the few-shot setting uses real-labeled data to select codes. The reason for the decreased performance lies in that the pseudo data of the co-reference task were of low quality and did not select effective task codes.

Please refer to Appendix A.3 for generated zero-label cases from CTR .

## 4.3 ABLATION STUDY

### 4.3.1 ARE MANUAL PROMPTS NECESSARY?

We are interested in whether CTR will be further improved when combined with discrete manual prompts. We conduct comparative experiments of CTR respectively with and without manual prompts. Specifically, for CTR without manual prompts, the inputs are a direct concatenation of multiple text fields, with compositional task representations appended in front of it. For CTR with manual prompts, the inputs are constructed by leveraging T0 prompts , with CTR placed in front.

The results are presented in Table 2. We report both the CTR results as well as its "upper-bound" results. It is noteworthy that the "upper-bound" results are post-hoc, which are obtained by the best code/prompt with observing the test task performance, and are merely given for estimating the potential of the two methods. We shall observe that adding additional discrete manual prompts does not improve the CTR performance as well as the "upper-bound". From the results, we shall conclude that (1) our CTR does not rely on manual prompts to obtain optimal performance. CTR can work as an alternative to the prompt-based methods. (2) By comparing the CTR performance and the CTR

upper-bound, there is still room to optimize the code-searching algorithm, and as a result to further improve the generalization performance.

| | Natural Language Inference | | | | | Sentence Completion | | | Co-reference | | WSD | Avg. |
|---|---|---|---|---|---|---|---|---|---|---|---|---|
| | RTE | CB | ANLI1 | ANLI2 | ANLI3 | COPA | Hella. | Story. | WSC | Wino. | WiC | |
| Our CTR | 80.51 | 87.50 | 33.40 | 34.40 | 33.80 | 92.00 | 27.50 | 90.10 | 56.58 | 49.40 | 62.50 | 58.88 |
| upper-bound | 81.95 | 89.29 | 38.77 | 39.00 | 40.60 | 94.23 | 42.81 | 94.20 | 73.86 | 60.00 | 61.76 | 65.13 |
| w/ Manual Prompt | 72.56 | 82.14 | 35.00 | 35.20 | 38.90 | 90.00 | 30.30 | 92.70 | 55.68 | 52.00 | 55.33 | 58.17 |
| upper-bound | 80.87 | 85.71 | 41.50 | 39.80 | 44.60 | 93.00 | 38.76 | 95.60 | 71.59 | 59.90 | 60.97 | 64.75 |

Table 2: Ablation study on manual prompts. It shows the results of our CTR respectively without and with manual prompts. The experiments are under the zero-label setting. The "upper-bound" results of each method are obtained by using the best code/prompt after observing test task performance and are merely given for estimating the potential of the method. Results show that additionally adding manual prompts does not necessarily improve performance, and the codebook learned by CTR can act as an alternative to previous manual prompts.

### 4.3.2 Training Objectives

Our CTR consists of three training loss items. Intuitively, the LM loss is optimized to predict the correct answer of the tasks. The embedding loss and the commitment loss are optimized to minimize the distance between the query embeddings and the CTR that are lookup from the codebook embeddings. To study the effectiveness of each loss item, we conduct an ablation study on them, and the results are shown in Table 3. Results show that removing either of the loss items will drastically hurt the zero-label performance, proving that either loss item is indispensable for the training of CTR .

| Loss Function | Natural Language Inference | | | | | Sentence Completion | | | Co-reference | | WSD | Avg. |
|---|---|---|---|---|---|---|---|---|---|---|---|---|
| | RTE | CB | ANLI1 | ANLI2 | ANLI3 | COPA | Hella. | Story. | WSC | Wino. | WiC | |
| Our CTR Loss | 80.51 | 87.50 | 33.40 | 34.40 | 33.80 | 92.00 | 27.50 | 90.10 | 56.58 | 49.40 | 62.50 | 58.88 |
| wo/ Commitment Loss | 76.53 | 76.79 | 33.30 | 32.60 | 38.00 | 85.00 | 33.50 | 74.70 | 64.77 | 49.90 | 54.39 | 56.32 |
| wo/ Embedding Loss | 74.01 | 75.00 | 33.70 | 33.20 | 34.30 | 75.00 | 25.40 | 86.00 | 61.36 | 54.30 | 50.00 | 54.75 |
| wo/ Commitment+Embedding | 79.06 | 76.79 | 30.90 | 33.70 | 35.70 | 84.00 | 30.50 | 85.90 | 57.95 | 55.60 | 50.00 | 56.37 |

Table 3: Ablation study on CTR loss function. Results show that by removing either of the loss items, i.e., commitment loss, embedding loss or both, the performance decreases to varying degrees.

### 4.4 In-Depth Analysis

### 4.4.1 Interpretability

Since CTR is trained to represent tasks with compositional codes, each code associated with one of the key aspects of tasks, CTR demonstrates a certain degree of interpretability. Table 4 presents examples of CTR that shows how each compositional task code is possibly interpreted.

Interestingly, it frequently occurred that tasks that share similar features indeed have the same compositional code. For example, the code 52 occurs in tasks that require extracting information from the given contexts, including samsum_* (summarization task), wiki_bio_* (structured data to text task), and paws_labeled_final_paraphrase_task (paraphrase generation) etc. Another example is that code 111 exists in most of the tasks that require generating long setences, including multi_news_* and samsum_* (both are summarization tasks).

We also validate these explanations on unseen test tasks. Results show that these possible explanations still hold. For example, the COPA task generates long answers given two candidate choices, and its compositional code has the code 111 (indicating sentence generation).

### 4.4.2 Controllability

CTR also demonstrates a certain degree of controllability. Specifically, by modifying one bit of the compositional task code, our CTR will exhibit a different task behavior. Table 5 shows several examples of controlling CTR. We shall observe that, given the same inputs, by simply changing one

| **Code:** 41 | **Explain: generate key words based on passages** |
|---|---|
| **TASKS:** | |
| wiki_hop_original_choose_best_object_interrogative_2 | wiki_hop_original_generate_subject_and_object |
| wiki_hop_original_choose_best_object_affirmative_3 | common_gen_sentence_to_concepts |
| common_gen_topics_from_the_sentence | |
| **Code:** 52 | **Explain: extract the main information of the article** |
| **TASKS:** | |
| samsum_Sum_up_the_following_dialogue | samsum_Summarize_this_dialogue |
| samsum_To_sum_up_this_dialog | samsum_Generate_a_summary_for_this_dialogue |
| samsum_Given_the_above_dialogue_write_a_summary | wiki_bio_key_content |
| wiki_bio_comprehension | wiki_bio_what_content |
| paws_labeled_final_paraphrase_task | |
| **Code:** 69 | **Explain: generate the answers given candidate choices** |
| **TASKS:** | |
| qasc_qa_with_separated_facts_1 | qasc_qa_with_separated_facts_2 |
| qasc_qa_with_separated_facts_4 | qasc_qa_with_combined_facts_1 |
| cosmos_qa_context_description_question_answer_text | cosmos_qa_description_context_question_answer_text |
| cosmos_qa_context_question_description_answer_text | cosmos_qa_no_prompt_text |
| social_i_qa_Show_choices_and_generate_answer | trec_fine_grained_LOC_context_first |
| trec_fine_grained_LOC | super_glue_copa_best_option |
| **Code:** 111 | **Explain: Generate long sentence** |
| **TASKS:** | |
| multi_news_summary_scenario | multi_news_summarize |
| multi_news_distill | multi_news_expand_(reverse_task) |
| samsum_Write_a_dialogue_that_match_this_summary | super_glue_copa_best_option |

Table 4: Examples of how compositional task codes can be possibly interpreted. We use a codebook size of 128. It shows the co-occurrence of tasks and codes. By analyzing common features shared by multiple tasks, we shall find that CTR embraces the advantage of interpretability.

bit of the compositional task code, the task behavior of CTR turns from DIALOGUE GENERATION to TOPIC CLASSIFICATION, from REVIEW RATING to SUMMARIZATION, from SENTIMENT ANALYSIS to TOPIC CLASSIFICATION, etc.

We shall conclude— (1) Since the inputs are randomly sampled from all tasks, some of which are quite different in data distribution, CTR performs well on them, proving that CTR indeed learns the ability to perform different tasks, instead of simply memorizing/overfitting to a certain dataset. (2) CTR is capable of switching between different tasks by simply changing one bit of the compositional code, proving that CTR effectively encodes the "task behavior" factor into the bits of the compositional code while disentangling other factors.

### 4.4.3  HOW CTR GENERALIZES TO NEW TASKS

To reveal how CTR essentially works, we use Figure 1 to explain how CTR generalizes from training tasks to unseen tasks such as COPA. During the training phase, CTR learns the compositional task code for each task, each code associated with certain features of the task. For example, code 69 is associated with many of the multiple-choice/reasoning-based QA tasks; code 111 is associated with most of the long-sentence-generation tasks, etc. For a new task, its task code is a re-arranged composition of the code existing in training tasks. If the new task shares the same code snippet as the training tasks, it inherits corresponding task features as well. In the example of Figure 1, a real case produced by our model, the new task COPA is performed by combining the abilities of reasoning-based QA, sentence generation, and multi-choice selection from training tasks. These observations are consistent with our motivation of learning CTR for cross-task generalization.

### 4.4.4  GENERALIZING TO NEW UNSEEN DOMAINS/CLASSES

Another interesting finding is that, for tasks with limited categorical classes, e.g., topic classification tasks, CTR is capable of generalizing to new unseen classes/domains. Table 6 shows several examples. Given different inputs that are randomly sampled from a large set of tasks and the compositional task code of AGNews, CTR not only predicts topic classes that are originally defined

| |
|---|
| **INPUT:** If Jim moves some particles of matter farther apart, what will happen to the rate at which they can pass vibrations on to nearby particles decrease or increase? Assuming that: When particles of matter are closer together, they can more quickly pass the energy of vibrations to nearby particles. |
| **TASK CODE:** [61, 96, 127, 113, 55, 107, 5, 87, _, 5] |
| **FILL 52 → DIALOGUE:** W: Hi, Jim. I'm Jim. I'm a physicist at the University of California, Berkeley. **FILL 37 → TOPIC CLASSIFICATION:** Science and Technology |
| **INPUT:** A historic victory for the separatist Parti Quebecois was marred by an attack on a victory rally for premier-elect Pauline Marois. A man opened fire during her victory speech, killing one person and critically wounding another, reports CTV. ... "I have convictions and I am going to defend them." Marois said during her victory speech. "There will be a referendum when the Quebec population wants a referendum." |
| **TASK CODE:** [31, _, 4, 113, 55, 107, 18, 87, 12, 63] |
| **FILL 68 → REVIEW RATING:** 1 star **FILL 93 → SUMMARIZATION :** One person was killed and another critically injured in a shooting at a victory rally for the separatist Parti Quebecois in Quebec City on Sunday. |
| **INPUT:** Paragraph: I've been here a few times and I like that it's right up the street. On Tuesday and Wednesday they have specials on manicures and pedicures which is $5 cheaper then the original price. |
| **TASK CODE:** [23, 90, 76, 15, 2, 79, _, 1, 119, 19] |
| **FILL 13 → SENTIMENT ANALYSIS:** Positive **FILL 5 → TOPIC CLASSIFICATION :** Business. |

Table 5: Examples of controlling the compositional codes. Given the input and a compositional task code, each time we modify one bit of the code, CTR begins to perform a different task.

| |
|---|
| **Predict with the code for AGNews (topic classification):** [23, 90, 76, 15, 2, 79, 39, 1, 119, 19] **Original AGNews classes:** {World, Sports, Business, Science and technology} |
| **INPUT:** Fears for T N pension after talks Unions representing workers at Turner Newall say they are 'disappointed' after talks with stricken parent firm Federal Mogul. **OUTPUT:** Business |
| **INPUT:** There are 10 apples on an apple tree. Three fall off. Now there are X apples. **OUTPUT:** Math problem |
| **INPUT:** Stuning even for the non-gamer This sound track was beautiful! It paints the senery in your mind so well I would recomend it even to people who hate game music! It would impress anyone who cares to listen! **OUTPUT:** Music |
| **INPUT:** Slack (2003) compares three groups that conducted biological research at Yale during overlapping periods between 1910 and 1970. Yale proved important as a site for this research. ... Hutchinson's example shows that new models for research groups are needed, especially for those that include extensive field research. **OUTPUT:** Ecology |

Table 6: Examples of CTR generalizing to new unseen classes/domains. The inputs are randomly selected from all tasks other than AGNews. The first case predict the same classes as AGNews defines, while the latter three cases predict new classes that are never seen within AGNews. It shows the codebook of CTR can generalize to new unseen classes/domains.

by AGNews, but also predicts new topic classes that never occur in AGNews. Please refer to Appendix A.1 and Appendix A.2 for more cases.

## 5 CONCLUSIONS

In this paper, we explore cross-task generalization from a new perspective—compositional generalization. We propose the Compositional Task Representations (CTR) method that learns a discrete compositional codebook for tasks and generalizes to new unseen tasks by forming new compositions of the task codes. For the inference of CTR , we propose two algorithms — Code Ensemble and Bitwise Search, respectively for zero-label and few-shot settings. Experiments demonstrate that our CTR significantly outperforms existing prompted-based methods on both zero-label and few-shot settings. Analysis of the learned compositional task codes proves that some of the CTR codes show certain degrees of interpretability and controllability.

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

# A   APPENDIX

## A.1   EXAMPLES OF AGNEWS

---

**QUESTION:** Fears for T N pension after talks Unions representing workers at Turner Newall say they are 'disappointed' after talks with stricken parent firm Federal Mogul.
**MODEL ANSWER (WITHIN THE ORIGINAL CLASSES):** Business

---

**QUESTION:** Prediction Unit Helps Forecast Wildfires (AP) AP - It's barely dawn when Mike Fitzpatrick starts his shift with a blur of colorful maps, figures and endless charts, but already he knows what the day will bring. Lightning will strike in places he expects. Winds will pick up, moist places will dry and flames will roar.
**MODEL ANSWER (WITHIN THE ORIGINAL CLASSES):** Science and technology

---

**INPUT:** How did serfdom develop in and then leave Russia?
**MODEL OUTPUT (OUT OF ORIGINAL CLASSES):** History

---

**INPUT:** There are 10 apples on an apple tree. Three fall off. Now there are X apples.
**MODEL OUTPUT (OUT OF ORIGINAL CLASSES):** Math problem

---

**INPUT:** the rock is destined to be the 21st century's new "conan" and that he's going to make a splash even greater than arnold schwarzenegger, jean-claud van damme or steven segal.
**MODEL OUTPUT (OUT OF ORIGINAL CLASSES):** Film

---

**INPUT:** E. D. Abbott Ltd - Abbott of Farnham E D Abbott Limited was a British coachbuilding business based in Farnham Surrey trading under that name from 1929. A major part of their output was under sub-contract to motor vehicle manufacturers. Their business closed in 1972.
**MODEL OUTPUT (OUT OF ORIGINAL CLASSES):** Company

---

**INPUT:** Stuning even for the non-gamer This sound track was beautiful! It paints the senery in your mind so well I would recomend it even to people who hate vid. game music! I have played the game Chrono Cross but out of all of the games I have ever played it has the best music! It backs away from crude keyboarding and takes a fresher step with grate guitars and soulful orchestras. It would impress anyone who cares to listen!
**MODEL OUTPUT (OUT OF ORIGINAL CLASSES):** Music

---

**INPUT:** E. D. Abbott Ltd - Abbott of Farnham E D Abbott Limited was a British coachbuilding business based in Farnham Surrey trading under that name from 1929. A major part of their output was under sub-contract to motor vehicle manufacturers. Their business closed in 1972.
**MODEL OUTPUT (OUT OF ORIGINAL CLASSES):** Company

---

**QUESTION:** Slack (2003) compares three groups that conducted biological research at Yale during overlapping periods between 1910 and 1970. Yale proved important as a site for this research. The leaders of these groups were Ross Granville Harrison, Grace E. Pickford, and G. Evelyn Hutchinson, and their members included both graduate students and more experienced scientists. All produced innovative research, including the opening of new subfields in embryology, endocrinology, and ecology, respectively, over a long period of time. Harrison's group is shown to have been a classic research school; Pickford's and Hutchinson's were not. Pickford's group was successful in spite of her lack of departmental or institutional position or power. Hutchinson and his graduate and postgraduate students were extremely productive, but in diverse areas of ecology rather than one focused area of research or the use of one set of research tools. Hutchinson's example shows that new models for research groups are needed, especially for those that include extensive field research.
**MODEL ANSWER (OUT OF ORIGINAL CLASSES):** Ecology

---

Table 7: Examples of CTR by the compositional code of AG_News. Given different inputs that are randomly sampled from a large set of tasks, and the compositional task code of AGNews, CTR not only predicts topic classes that are originally defined by AGNews, but also predicts new topic classes that never occur in AGNews .

A.2   EXAMPLES OF DBPEDIA_14

---

**QUESTION:** Federation of International Trade Associations - The Federation of International Trade Associations (FITA) based in Reston Virginia and New York New York USA was founded in 1984. It fosters international trade by seeking to strengthen the role of associations in the United States Mexico and Canada. FITA is the strategic partner of the United States Commercial Service for e-commerce. company, educational institution, artist, athlete, office holder, mean of transportation, building, natural place, village, animal, plant, album, film or written work
**MODEL ANSWER (WITHIN THE ORIGINAL CLASSES):** Company

---

**QUESTION:** Furian knife In this movie sequel, Vin Diesel returns as Riddick, an escaped convict with a price on his head. Riddick has been hiding on a snow planet for the last five years, when a group of mercenaries try to capture him. Riddick returns to the planet Helion, and finds out that his friend Jack, is in prison on Crematoria, a very hot planet. While on Helion, the planet is invaded by the Necromongers, led by Lord Marshal (Colm Feore), who wants to rule the universe. Riddick is captured by the mercs and flown to the same unsavoury and possibly illegal prison Kyra is in. Turns out Jack is now known as Kyra (Alexa Davalos) and is tough as nails.
**MODEL ANSWER (WITHIN THE ORIGINAL CLASSES):** Film

---

**QUESTION:** lila abu-lughod's: - name - alma mater - website - known for - birth date - employer - nationality - occupation Bio: lila abu-lughod -lrb- born 1952 -rrb- is an american with palestinian and jewish ancestry who is professor of anthropology and women's and gender studies at columbia university in new york city. a specialist of the arab world, her seven books, most based on long term ethnographic research, cover topics from sentiment and poetry to nationalism and media, from gender politics to the politics of memory.
**MODEL ANSWER (WITHIN THE ORIGINAL CLASSES):** Educational Institution

---

**INPUT:** when he dug his massive body into the batter's box tuesday night at legends field during an exhibition game against the new york yankees, adam dunn not only wasn't hitting his weight, he wasn't hitting the weight of an UNK bag of baseballs.
**MODEL OUTPUT (OUT OF ORIGINAL CLASSES):** Sports

---

**INPUT:** We know that, thanks to our DNA, each of us is a little bit different. Some of those differences are obvious, like eye and hair color. Others are not so obvious, like how our bodies react to medication. Researchers are beginning to look at how to tailor medical treatments to our genetic profiles, in a relatively new field called pharmacogenomics. Some of the biggest breakthroughs have been in cancer treatment. **MODEL OUTPUT (OUT OF ORIGINAL CLASSES):** Science

---

**INPUT:** how did athenians make money? Other Greek cities set up democracies, and even though most followed an Athenian model, none were as powerful, stable, nor as well-documented as that of Athens.
**MODEL OUTPUT (OUT OF ORIGINAL CLASSES):** City

---

**INPUT:** Great food, portions could be smaller. A little pricey for Middleton. Would try somewhere else next time but would also not hesitate to return
**MODEL OUTPUT (OUT OF ORIGINAL CLASSES):** Restaurant

---

**INPUT:** Facts: - name: mr. marcus - spelling: us - caption: mr. october 2007 marcus at a porn star karaoke event, - height: 5 8 - alias: mister marcus, mr. marquis - image: mr marcus, 2007.jpg - birth date: 4 september 1970 - birth name: jesse spencer - weight: 200 lb kg on - ethnicity: black - birth place: pomona, california, usa - hair color: black - number of films: 1,782 -lrb- per iafd -rrb- as a performer & 36 as a director - eye color: brown mr. marcus.
**MODEL OUTPUT (OUT OF ORIGINAL CLASSES):** Actor

---

Table 8: Examples of CTR by the compositional code of DBpedia_14. Given different inputs that are randomly sampled from a large set of tasks, and the compositional task code of DBpedia_14, CTR not only predicts topic classes that are originally defined by DBpedia_14, but also predicts new topic classes that never occur in DBpedia_14 .

### A.3 Examples of Zero-Label Tasks

| |
|---|
| **COPA Task Code:** [111,7,69,15,2,79,39,1,119,19] |
| **INPUT:** The woman retired. so... - She received her pension. - She paid off her mortgage.
**OUTPUT:** She received her pension. |
| **INPUT:** The girl found a bug in her cereal. so... - She poured milk in the bowl. - She lost her appetite.
**OUTPUT:** She lost her appetite. |
| **Winogrande Task Code:** [5,21,76,15,2,79,25,1,119,19] |
| **INPUT:** Sarah was a much better surgeon than Maria so _ always got the easier cases. Sarah or Maria?
**OUTPUT:** Sarah |
| **INPUT:** Terry tried to bake the eggplant in the toaster oven but the _ was too big. eggplant or toaster?
**OUTPUT:** eggplant |
| **StoryCloze/2016 Task Code:** [23,68,54,10,2,79,25,103,120,44] |
| **INPUT:** Sam loved his old belt. He matched it with everything. Unfortunately he gained too much weight. It became too small. - Sam went on a diet.- Sam was happy.
**OUTPUT:** Sam went on a diet. |
| **INPUT:** Larry bought a new motorcycle. He was excited to look cool. The first time he tried riding it he dropped it. He hurt his leg and had to go to the hospital. - Larry loved going to the hospital.- Larry became careful.
**OUTPUT:** Larry became careful. |
| **Winograd Schema Challenge Task Code:** [5, 68, 4, 10, 8, 22, 18, 124, 12, 97] |
| **INPUT:** Jane gave Joan candy because she was hungry. Jane was hungry. Yes or no?
**OUTPUT:** No |
| **INPUT:** Joe Joe's uncle can still beat him at tennis, even though he is 30 years older. Joe is 30 years older. Yes or no?
**OUTPUT:** No |

Table 9: Examples of generalizing to tasks of unseen types. It shows cases of the CTR code for zero-label test tasks, and corresponding generated examples.

### A.4 Component Architecture

| Encoder-Decoder | Natural Language Inference | | | | | Sentence Completion | | | Co-reference | | WSD | Avg. |
|---|---|---|---|---|---|---|---|---|---|---|---|---|
| | RTE | CB | ANLI1 | ANLI2 | ANLI3 | COPA | Hella. | Story. | WSC | Wino. | WiC | |
| CTR: Linear - Linear | 80.51 | 87.50 | 33.40 | 34.40 | 33.80 | 92.00 | 27.50 | 90.10 | 56.58 | 49.40 | 62.50 | 58.88 |
| Linear - None | 77.98 | 82.14 | 32.40 | 33.30 | 33.60 | 86.00 | 29.43 | 89.30 | 55.68 | 57.70 | 50.00 | 57.05 |
| Linear - MLP | 77.26 | 80.36 | 32.40 | 34.00 | 35.60 | 84.00 | 26.86 | 84.10 | 54.55 | 55.70 | 50.31 | 55.92 |
| Linear - Transformer | 77.98 | 80.36 | 32.70 | 31.80 | 35.70 | 88.00 | 27.72 | 89.10 | 54.55 | 58.00 | 49.84 | 56.89 |
| Linear - RNN | 78.70 | 80.36 | 33.80 | 32.50 | 36.30 | 87.00 | 30.70 | 87.30 | 55.68 | 57.20 | 56.11 | 57.79 |
| MLP - Linear | 79.06 | 71.43 | 32.40 | 33.40 | 35.20 | 88.00 | 30.25 | 89.70 | 60.23 | 55.70 | 49.84 | 56.84 |
| MLP - None | 78.34 | 76.79 | 32.90 | 33.30 | 33.80 | 87.00 | 30.08 | 90.60 | 52.27 | 56.60 | 50.31 | 56.54 |
| MLP - MLP | 76.53 | 76.79 | 33.20 | 33.30 | 34.30 | 84.00 | 28.14 | 90.90 | 52.27 | 55.00 | 50.47 | 55.90 |
| MLP - Transformer | 78.70 | 75.00 | 32.40 | 33.00 | 33.90 | 82.00 | 28.70 | 91.00 | 56.82 | 55.70 | 50.16 | 56.12 |
| MLP - RNN | 79.78 | 73.21 | 33.20 | 33.70 | 34.50 | 85.00 | 28.64 | 90.90 | 57.95 | 56.90 | 50.00 | 56.71 |

Table 10: Ablation study on different architectures of the CTR encoder and decoder. All experiments are conducted under the zero-label setting. We experiment two of the encoder, respectively the linear net and the multi-layer perceptron (MLP), and five alternatives of the decoder, including the linear net, bidirectional RNN, Transformer, MLP and removing the decoder (None). Results show that the simple "Linear - Linear" combination achieves the best performance.

Table 10 shows the zero-label results when using different architecture of the encoder and the decoder. We experiment two of the encoder, respectively the linear net and the multi-layer perceptron (MLP), and five alternatives of the decoder, including the linear net, bidirectional RNN, Transformer, MLP and removing the decoder (None). Results show that simply using a linear network for both the encoder and the decoder performs best, while complicated architectures, e.g., Transformer and MLP, yield poor performance. This could because that the learning of codebook embeddings

as well as the code mapping does not necessarily need complicated transformation. When using complicated architectures, it instead increases the difficulties of the learning process.

## A.5   CODEBOOK SIZE AND CTR LENGTH

There are two critical hyper-parameters for CTR —the codebook size and the CTR length. We conduct experiments of different selection of the two hyper-parameters, and see how it influence the performance of zero-label generalization. Table 11 presents the results. We shall observe that when the codebook size decreases below 64, the performance decreases to a large degree. We conjecture that it is due to the capacity of a codebook size below 64 is not sufficient for representing the multiple aspects of the tasks. In addition, experimenting with a CTR length of 10 generally outperforms those with larger CTR length.

| Codebook Size | CTR Length | Natural Language Inference | | | | | Sentence Completion | | | Co-reference | | WSD | Avg. |
|---|---|---|---|---|---|---|---|---|---|---|---|---|---|
| | | RTE | CB | ANLI1 | ANLI2 | ANLI3 | COPA | Hella. | Story. | WSC | Wino. | WiC | |
| 128 | 10 | 80.51 | 87.50 | 33.40 | 34.40 | 33.80 | 92.00 | 27.50 | 90.10 | 62.50 | 49.40 | 56.58 | 58.88 |
| 128 | 20 | 77.62 | 80.36 | 33.10 | 34.40 | 38.50 | 87.00 | 31.60 | 90.90 | 54.55 | 50.90 | 55.80 | 57.70 |
| 64 | 10 | 75.81 | 73.21 | 31.90 | 31.70 | 36.70 | 89.00 | 26.90 | 65.40 | 48.86 | 52.50 | 54.70 | 53.34 |
| 64 | 20 | 79.06 | 82.14 | 36.80 | 31.70 | 38.50 | 85.00 | 30.86 | 92.00 | 56.82 | 53.20 | 50.47 | 57.87 |
| 48 | 10 | 77.26 | 78.57 | 35.30 | 33.30 | 38.80 | 88.00 | 33.54 | 66.10 | 60.23 | 50.00 | 53.45 | 55.87 |
| 48 | 20 | 79.42 | 60.71 | 33.80 | 32.50 | 33.20 | 81.00 | 29.00 | 92.50 | 50.00 | 48.00 | 53.92 | 54.01 |

Table 11:   Ablation study on different codebook size and CTR length. All experiments are conducted under the zero-label setting. We experiment three of the codebook size, respectively 128, 64 and 48, and two of the CTR length, including 10 and 20. Results show that "128 - 10" combination achieves the best performance.

## A.6   PERFORMANCE SENSITIVITY TO DIFFERENT SELECTION OF CODES

We want to verify whether different selections of codes will affect the performance of the model. Intuitively, if a code is merely noise that gets ignored by the model eventually, the model will not sensitive to different selections of codes and vice versa. Therefore, we randomly sampled several codes of training tasks and evaluate their performance on different test tasks. The results are shown in Table 12. As shown in the table, the model is highly sensitive to a different selection of codes. For example, the code of task amazon polarity_user_satisfied can perform well on RTE and significantly better than the performance of code of task cnn_dailymail_3.0.0_news_card_view.

| Test Task | Code of Train Task | Results | Min./Max. |
|---|---|---|---|
| CB | xsum_read_below_DOC_write_abstract
quoref_Guess_Title_For_Context
imdb_Writer_Expressed_Sentiment
gigaword_write_a_title_for_this_sentence | 35.83
41.79
52.74
42.50 | 35.83/52.74 |
| RTE | amazon_polarity_user_satisfied
kilt_tasks_hotpotqa_final_exam
cnn_dailymail_3.0.0_news_card_view
wiki_hop_original_explain_relation | 75.13
72.74
49.60
49.36 | 49.36/72.74 |
| WSC | cosmos_qa_context_description_question_answer_text
yelp_review_full_format_score
cos_e_v1.11_question_description_option_text
xsum_DOC_write_summary_of_above | 52.74
63.51
57.90
53.19 | 52.74/63.51 |
| COPA | cos_e_v1.11_rationale
quarel_logic_test
wiki_hop_original_choose_best_object_interrogative_2
adversarial_qa_dbert_based_on | 55.95
81.92
59.05
81.40 | 59.05/81.92 |

Table 12: Performance sensitivity to different selections of codes.

## A.7 EXPERIMENTAL DETAILS

For the data preprocessing of all experiments, to balance the number of data for different tasks, we restrict the maximum data examples for each training task to be 50,000, which empirically yields better results.

Training details of each baseline method under the zero-label setting are illustrated as follows.

**T0-Large** Based on T5-Large-LM-Adapted, it performs multi-task training for 10000 steps. We set the maximum length of input and target sequences to 384 and 32 respectively. We use the Adam optimizer with a learning rate of 1e-4, a dropout rate of 0.1, and a batch size of 1024. Following T0 Sanh et al. (2022), we use the same task prompts from PromptSource(Bach et al., 2022). We report the average accuracy of multiple prompts for each test task. Note that our reproduced T0-Large results are much better than those reported in the original paper Sanh et al. (2022), which sets a much stronger baseline for comparison. We report the average accuracy of prompts for each test task. We believe our baseline is well-optimized. Because the performance on test tasks is comparable to the results of T0-3B reported in Sanh et al. (2022), even our baseline only contains 770M parameters.

**Self-Training** For a fair comparison, we randomly sample 32 unlabeled data for self-training, which is the same as CTR. The Self-Training method trains from the T0-Large with these pseudo-labeled examples for 5 epochs and reports average performance over prompts. For the training, we use a batch size of 32 and the Adam optimizer with a learning rate of 1e-4.

**Manual-Code** In practice, we manually label a set of artificially-designed features for each task, including the number of input fields of the task, whether it requires reasoning, whether it includes options into inputs, and whether it is a classification task, etc. Each task can be represented as an artificially-defined discrete feature vector, each dimension associated with one of the aspects. Manual-Code follows exact the same training recipe as the second training phase of our CTR. Training details are presented in Section 4.1.3.

**Zero-Label Prompt Selection (ZPS)** (Liao et al., 2022a) For each task, we use 32 unlabeled data for producing pseudo-labeled data. Finally, we report the accuracy of the selected prompt.

For the few-shot setting, we consider the following five baseline methods. All few-shot baselines are based on our reproduced T0-Large.

**Model Tuning** We use the Adam optimizer with a batch size of 256 and a learning rate of 1e-4. We combine all training data of the test tasks for training. The maximum training step is 100. We use a validation set for model selection. Finally, we report the average accuracy of the selected best checkpoint.

**Prompt Tuning** (Lester et al., 2021) We use the Adam optimizer with a batch size of 128 and a learning rate of 0.05. We combine all training data of the test tasks for training. The maximum training step is 100. The length of continuous prompts for each task is 20. We use a validation set for model selection. Finally, we report the average accuracy of the selected best checkpoint.

**GPS** (Xu et al., 2022) We follow hyper-parameters reported in the original paper. Specifically, we run the GPS for 6 steps. At each step, new prompts are generated by a T5-xxl-lm-adapted model.

**GRIPS** In our experiments, we set max patience $P = 2$, candidate $l = 5$, and step $m = 5$. For the rest of the hyper-parameters, we follow the original GRIPS paper(Prasad et al., 2022).

**Black-Box Tuning (BBT)** (Sun et al., 2022) In practice, we use the Adam optimizer with a learning rate of 0.05. For each test task, we train the soft prompt for 200 steps. We set the prompt length $L = 10$, subspace dimension $d = 500$, and cma_budget 1000. For the rest of the hyper-parameters, we follow the original BBT(Sun et al., 2022).

