# OpenReview forum: "Compositional Task Representations for Large Language Models"
_ICLR.cc/2023/Conference — ICLR 2023 poster_

### Official Review · Reviewer_zu2t · 2022-10-15

**Confidence:** 5
**Correctness:** 3
**Technical Novelty And Significance:** 3
**Empirical Novelty And Significance:** 3
**Recommendation:** 6

**Clarity, Quality, Novelty And Reproducibility:**

Clarity is excellent.

Quality is quite good, with the lack of details on hyper-parameter selection and baselines being the main issue.

Novelty is not an issue.

Reproducibility is a bigger issue. Not a lot of clarity on hyper-param tuning, baselines and the value of some parameters (such as /beta) are not given.

**Strength And Weaknesses:**

Pros:
* The paper is well-written and well-structured. The approach is well explained, related work is presented clearly and the experimental setup is clear.
* The set of baselines is comprehensive and very helpful in assessing the value of the method presented. Having both the zero-shot and few-shot settings is also helpful.
* The method achieves good results overall, although there may be some issues with measurement (see Cons section).
* Both ablations are interesting and helpful.
* The analysis section is rich and sheds light in some of the properties and benefits of CTR.

Cons:
* My main concern is on the experimental setup, especially hyper-parameter tuning. For this setup, the proper way to do hyper-parameter tuning is not always obvious. Table 10 and 11 shows the authors have tried several configurations/options (at least 15) and this paper presents the best one. Given the wide variation on results and limited data of some tasks, the authors need to clarify how they performed hyper-parameter tuning for their method as well as the baselines. This will ensure the comparison in Table 1 is fair.
* There needs to be more information on the experimental setup for the baselines. For instance, the authors introduce the "Manual-code" baseline but very little detail is given. For the self-training baseline, we do not know the experimental details at all. In general this section is lacking and makes the comparison of Table 1 harder.
* There are details missing that harm reproducibility. For instance, the value of the beta parameter selected for the loss is not given, let alone how selection was done.
* I would comment on the text that there is wide variability across tasks for Table 1. For instance, Discarding the CB outperformance the method is very similar to prior ones on the zero-shot setting. Given the aforementioned sparsity of details on hyper-parameter selection and baselines this is an even bigger issue.
* Some further ablations would be interesting:
  * How does the method perform if fewer tasks are provided during training (currently 319).
  * Does the order of the latent codes matter for performance? At inference time, how does performance deteriorate if you shuffle the order of the codes selected for a task.
  * Visualizing the task embeddings / the codes may be interesting.
* Minor: Typo p8 combing -> combining

**Summary Of The Paper:**


This paper introduces Compositional Task Representations, an approach to train language models to achieve strong zero-shot and few-shot performance on unseen tasks without the need to rely on prompts at test time.

Given a pre-trained LM, CTR is trained on a variety of tasks. For each task, a vector is initialized and trained. An encoder then produces a sequence of query vectors from this vector, These query vectors are used to retrieve a sequence of codes from a global set of trainable codes. Those codes are given as input to the LM, serving as a task representation meant to replace the prompt. At inference time on a new task, the authors search for a good composition of codes for the new tasks using unlabelled examples.

The authors compare CTR to a comprehensive set of recent baselines on both the zero-shot and few-shot settings on the T0 benchmark. They achieve strong results overall on both these settings, outperforming other methods. They present ablations on the use of prompts at test time for CTR and on the importance of sub-components of their loss. Finally, there is a rich analysis section showing the interpretability, controllability and generalization ability of CTR.


**Summary Of The Review:**

This is a well-written paper that introduces an interesting method to train LM to have strong zero and few-shot capabilities. In addition to good results, the method offers nice advantages in terms of controllability and interpretability. The main issue with this paper is in the lack of clarity on hyper-parameter selection and details of the baselines. Given the results seem to be quite sensitive to these minutes details, this casts some doubt into the claimed performance.

---

> ### Author Response · Authors · 2022-11-19
> **Response to Reviewer zu2t**
>
> We thank the reviewer for the valuable feedback. We would like to clarify a few points as follows.
> 1. We have added detailed descriptions to our paper. Since it is costly to conduct a single experiment, we did not perform large-scale hyper-parameter tuning. For hyper-parameters shared by T0, we directly followed exactly the same training recipe as T0. For unique hyper-parameters such as CTR length and codebook size, we determine values according to practical considerations (i.e., empirical values).
> 2. Same as Response #1, we have also added detailed illustrations in terms of baselines in our paper. Please refer to Section A.4.
> 3. We have added experimental details to our paper. For reproducing the results, we will also release our code and scripts.
> 4. Our Code Ensemble method uses pseudo-labeled data to select test codes so it does not show a clear advantage on the tasks that T0 cannot perform well, such as ANLI or Winogrand. When labeled data is given(few-shot setting), the improvement compared to T0-Large is consistent and our method shows obvious superiority on most datasets in the few-shot setting. For concerns about hyper-parameters and details of baselines, please see Response #1.
> 5. Response to #4: Very interesting point. Currently, we experimented with several critical aspects, which determine CTR holds. We do agree that adding these ablation studies will make CTR a more solid work. We are now conducting experiments and will add these results to a future version.
> 6. Thank you for pointing out the typo. We have corrected it.

---

> > ### Comment · Reviewer_zu2t · 2022-11-19
> > **Thanks for your answer, further clarification needed on hyperparameters.**
> >
> > Hi,
> >
> > Thank you for your answer. For hyper-parameters, my question was specifically about parameters of the method (encoder and decoder architecture Table 10, codebook size and length Table 11). These parameters impact performance significantly and "we determine values according to practical considerations (i.e., empirical values)" does not strike me as a clear recipe.
> >
> > It is not immediately clear to me what practical considerations on empirical values made you choose that Linear-Linear was better than e.g Linear-RNN or that Codebook size 128 and CTR length 10 was better than 128 and 20 aside from the results of the ablations themselves in Table 10 and 11. These changes impact by 1+ points your results so they are significant. I do not think selecting given downstream performance works as it leads to over-estimating the performance of the model.

---

> > > ### Author Response · Authors · 2022-11-24
> > > **Response to Reviewer zu2t**
> > >
> > > Hi. Thank you for the response.
> > >
> > > First of all, "empirical values" mean that we performed preliminary experiments.  We selected the hyper-parameters through post-hoc observations over the test sets, and then fixed them in the following experiments, **which is exactly the same as the baseline T0**.
> > >
> > > We do agree that it would be somewhat problematic (e.g., the risk of overestimation). However, there are several rationales and rationalities---
> > > 1. We used exactly the same strategy as the T0 baseline for hyper-parameter determination, both sharing the same pros & cons.
> > > 2. It would be unfair to compare if we split an extra validation set while T0 does not, since in such cases both our method and baselines will use completely different train data.
> > > 3. We currently follow the same task setting as T0. The fact that the T0 task set is very imbalanced makes it not feasible to split a validation set from the training set for hyper-parameter selection.  For example, the summarization tasks account for 58.92% of all data, while the closed book QA type accounts for 0.28% of all data.
> > >
> > > To guarantee a zero-shot setting, we have to split an entire task type as the validation set. Our preliminary experiments show that it would make a big difference if we split different task types as the validation set. The imbalanced tasks and splits will lead to biased results.
> > >
> > > So it would be a better solution to expand the data and tasks and to split a validation set for hyper-parameter selection, and we will  leave it to future work! We appreciate your constructive feedback, and will further improve the paper in the future revision.

---

> > > > ### Comment · Reviewer_zu2t · 2022-11-28
> > > > **Thanks for your clarification, still take some issue with this tuning method.**
> > > >
> > > > Thanks for the clarifications. I understand that this mirrors T0 in its idea. However, T0 runs one single full-sized model for a given task split (T0, T0+ or T0++). The ablations there are on the splits themselves (or the prompts, but it is not extensive and motivation is clearer). In contrast, your ablations are on the architecture. You run at least 15 configurations, performance varies significantly and the chosen numbers are the best. It is much more likely that you overestimate your numbers.
> > > >
> > > > I am aware that there is no good way this can be corrected a posterior and that carving out another validation set / different splits is not practical.

---

> > > > > ### Author Response · Authors · 2022-12-01
> > > > > **Response to Reviewer zu2t**
> > > > >
> > > > > We thank the reviewer for the additional discussion.
> > > > >
> > > > > We do agree that the current method of choosing hyper-parameters could be exposed to the risk of over-estimation, for both baselines and ours.  It still remains an open question of how to choose hyper-parameters without any prior of the target tasks. We will continue working on this problem to explore better solutions.
> > > > >
> > > > > Thank you for pointing out the valuable question.

---

### Official Review · Reviewer_xT8s · 2022-10-22

**Confidence:** 3
**Correctness:** 3
**Technical Novelty And Significance:** 3
**Empirical Novelty And Significance:** 3
**Recommendation:** 8

**Clarity, Quality, Novelty And Reproducibility:**


The paper is well-written.

Questions:
- Sometines, performance in the zero-labeling setting is better than the few-shot setting. I am wondering how much it is sensitive to the selection of codes. Plus, what is the scale of these data.
-- Another point is that, in some scenarios, we do have some annotated data. Can CTR also benefit from many annotated  examples?



**Strength And Weaknesses:**

# Strength
- The idea is useful and interesting
- the models provide some interpretability, controllability, and generilization

# Weaknesses
- have not found any significant one


**Summary Of The Paper:**

This paper proposes a prompt-based architecture to improve the performance of large language models on zero-label learning and few-shot learning. The authors employ multi-task training to learn a discrete, compositional codebook, which is for mapping the given task to task types and then generate a task-related prompt for LLM inference.

Learning a discrete latent codebook to enhance the interpretability of prompts is impressive, and the experiments are comprehensive and convincing.


**Summary Of The Review:**

The idea is useful and interesting. I generally like this idea. I recommend acceptance.

---

> ### Author Response · Authors · 2022-11-19
> **Response to Reviewer xT8s**
>
> We thank the reviewer for the valuable feedback.
> 1. Response to #1:
>     * In terms of sensitivity to the selection of codes, we have added additional experimental results in the Appendix. Results show that the zero-shot performance is quite sensitive to different codes (>20 points), proving that code selection is of vital importance.
>     * For the scale of the data, there're 319 tasks and 4 million train data. The CTR length is 10 and the codebook size is 128, resulting in a code space of 10^128 size. Though large code space, we provide algorithms that enable efficient search test codes in Section 3.3.
> 2. Very good point. We are now conducting experiments with full-labeled data and will add results under full-finetuning tasks in the future version.

---

### Official Review · Reviewer_FsjZ · 2022-10-24

**Confidence:** 3
**Correctness:** 3
**Technical Novelty And Significance:** 3
**Empirical Novelty And Significance:** 2
**Recommendation:** 6

**Clarity, Quality, Novelty And Reproducibility:**

Technical details in section 3.3 need to be explained with math/pseudocode and better clarity.
* The zero-label setting is not clear to me. The problem formulation states “given a new task along with a set of unlabelled data”. What are these unlabelled instances? How are they relevant to the task?
* “gives predictions that deviate from a uniform distribution” - Does this mean the proposed approach only applies to classification tasks?
* In the few-shot learning setting, the authors mention an algorithm which iteratively changes a single bit of code, but do not discuss further details.

Several aspects of the baselines were unclear/vague. I couldn’t gauge whether the comparison is fair due to these missing details.
* “T0 reports the average results over multiple prompts” - What are these prompts? Are they natural language prompts?
* Model Tuning baseline - Is this model trained on the same training tasks as the proposed method?
* Does CTR use any instructions?

Table 1: What are the evaluation metrics for each task?

How were the results in Table 5 obtained? Some results do not look very convincing (e.g. the dialog example in the first instance).

Other comments
* FIgure 2 has poor readability, fonts are too small
* Sec 3.1 says that the code embeddings are optionally passed through a decoder, but doesn’t explain what this means.


**Strength And Weaknesses:**

Pros
* Viewing task representations as compositional codes is an interesting perspective
* Proposed method offers some degree of interpretability and controllability in task learning

Cons
* The problem settings and proposed approach  are vaguely described.
* Several aspects of the experimental setting were unclear which makes it hard to interpret the results.


**Summary Of The Paper:**

This paper proposes a method for multitask learning of NLP tasks with compositional codes. Given a task id, a sequence of discrete codes are obtained. Code vectors obtained from these codes using an embedding table are fed as input to the language model as a task representation. The codebook and the language model are jointly trained with a language modeling loss and additional losses to enforce consistency between task ids and corresponding codes. Results show that the proposed approach performs better than prior methods in zero-label and few-shot settings. In addition, the trained control codes offer some degree of interpretability and controllability.

**Summary Of The Review:**

Updating score to 6 post author response.

The authors propose an interesting approach to learn task representations as compositional codes. However, due to missing details about the proposed approach, experimental setup and baselines it is difficult to gauge the impact of the approach. In addition, the claims about interpretability and controllability are not entirely convincing. See more detailed comments below.

---

> ### Author Response · Authors · 2022-11-19
> **Response to Reviewer FsjZ**
>
> We thank the reviewer for the insightful feedback. We would like to clarify a few points.
> 1. To clarify the problem setup and experimental setting, we have added detailed information.
>     * For the zero-label setting, the unlabeled data of the task equals the input of the task. It is noteworthy that the inputs naturally exist as long as we perform the evaluation. So it requires no extra effort to acquire data.
>     * Our proposed method can be applied to both classification and generation tasks. Similar to T0, CTR also formulates each task into the "input-target" format. Besides, the LLM of CTR uses an encoder-decoder architecture, which enables predicting targets conditioned on inputs. In the context of "gives predictions that....", the predictions can also be the targets of generation tasks.
>     * We have supplemented more detail about the Bitwise Search method in Section 3.3 to ensure the readers can clearly understand the method. Overall, we propose the Code Ensemble method for the Zero-Label setting and Bitwise Search for the few-shot setting. In table 1, we report the result of CTR under the Zero-Label Setting, where the selected code is derived by the Code Ensemble method. The last column of Table 1 is the result of CTR under the few-shot setting, where the selected code is derived by the Bitwise Search method. We also discuss the performance in Section  4.2.
> 2. Response to "Several aspects of the baselines were unclear....":
>     * The baseline T0 use prompts provided PromptSource. They are all natural language prompts, which are handcrafted by humans.
>     * Yes, The model tuning baseline is also trained on the same training tasks as ours.
>     * CTR did not use any instructions or prompts. Specifically, for CTR, the input texts are a direct concatenation of multiple text fields, with compositional task representations appended in front of it. In addition, we have an ablation study in Section 4.3.1, showing that adding manual instructions or prompts did not help CTR.
> 3. For Table 1, the evaluation metrics for each task are accuracy.
> 4. For Table 5, we randomly sampled a task code from training tasks and change one bit of the code to another one, and then we observe whether the model begin to perform a different task. Note a certain code can only appear in the fixed position of the task code and we only try the code that shows some kind of interpretability. Therefore the search space is small. Besides, we have to admit only one part of the codes is interpretable and the other part can not be understood by humans.
> 5. Response to "Other comment":
>     * Thank you for pointing out that. We have adjusted the font size in Figure2.
>     * We experimented with the CTR decoder in terms of different architectures (including None, i.e., it means directly feeding the CRT into LLM without passing through a decoder). All strategies work but yield various performances. We have adjusted the word to avoid misleading.

---

> > ### Comment · Reviewer_FsjZ · 2022-11-22
> > **Thank you for the response**
> >
> > Thank you for the response.
> >
> > Regarding the 'zero label setting', I do not agree that it requires no extra effort to acquire task-relevant inputs. This is a non-trivial assumption that needs to be clearly described and motivated in the paper.
> >
> > If the method is applicable to generation tasks, how does one measure if “predictions deviate from a uniform distribution”?
> >
> > Thank you for sharing additional details about the baselines.
> >
> > I recommend the authors to clearly describe and motivate the experimental setup as well as the proposed approach in their next revision.

---

> > > ### Author Response · Authors · 2022-11-24
> > > **Response to Reviewer FsjZ**
> > >
> > > Thank you for your valuable feedback.
> > >
> > > (1) Allow me to explain the setting. Perhaps we are familiar with the zero-shot setting, where the test tasks are completely unseen before evaluation. The main difference between the zero-shot setting and zero-label setting lies in that the zero-label setting will additionally make use of the input data $x$ of the test tasks. Providing input $x$ is an indispensable process, otherwise, it would be impossible to perform the evaluation.
> > > Note that our method does NOT make any requirements to the distribution of input data $x$. On the contrary, it is what input data distribution the user has provided, that defines what the zero-label task is. The ultimate goal of our method is to generalize to any zero-label tasks the users could provide. So it would not be problematic if we use the input data.
> > > Compared to the zero-shot setting, the zero-label setting is a more practical and feasible setting, since it allows real-time input data of any distribution, and makes an adaption to the coming test task accordingly.
> > >
> > > (2) Thank you for pointing out the limitation of presenting "predictions deviate from a uniform distribution".
> > > Indeed, the ``uniform distribution'' assumption is practical only for classification tasks. However, it is worth emphasizing that a similar idea can also be adapted to generation tasks only with slight modifications. For example,
> > > 1. We can use heuristic strategies to filter out low-quality codes (e.g., generations are too short, too long, or lack necessary word overlap between generations and inputs).
> > > 2. We generate pseudo targets on the unlabeled data by ensembling the logits obtained from a set of code candidates. Specifically, at each step of decoding, we predict the token through ensembled logits.
> > > 3. We select the code with the highest BLEU/ROUGE score.
> > > To follow the T0 benchmark, we currently only validate CTR on classification tasks. But we will leave the verification on generation tasks to future work!
> > >
> > > (3) We will definitely further improve relevant details in the next revision.

---

> > > > ### Comment · Reviewer_FsjZ · 2022-11-28
> > > > **Follow-up response**
> > > >
> > > > Thank you for the response.
> > > >
> > > > I understand what the authors refer to as the 'zero-label setting', but I don't think that this is a 'natural assumption' as the authors argue. I also do not agree it is a 'more practical and feasible setting' compared to zero-shot setting. This setting assumes that the target task's input distribution is known. The authors needs to analyze the impact of the number of unlabeled task-relevant inputs on the final performance.
> > > >
> > > > That said, the author response convinced me to raise my score. However, I still believe that the authors need to motivate their evaluation setting better and improve clarity.

---

> > > > > ### Author Response · Authors · 2022-12-01
> > > > > **Response to Reviewer FsjZ**
> > > > >
> > > > > We thank the reviewer for the insightful feedback. We would like to address a few points as follows.
> > > > >
> > > > > (1) To clarify, during the training process, the input distribution of target tasks is completely unknown and unused. The inputs of target tasks are only used during the inference process. Therefore, under the zero-label setting, it does not violate our requirements that the target tasks should be invisible during model training.
> > > > >
> > > > > (2) During the inference, the input distribution of a target task indeed has an impact on the selected code as well as the ultimate performance of the target task. We humbly take the advice to further study how the number of unlabeled inputs influences final performance.  Empirically, in all our experiments, we use a number of 32 unlabeled inputs for each target task, and all results prove that it is sufficient to select effective codes.
> > > > >
> > > > > Besides, we are also conducting experiments to use PLMs to generate inputs of any distribution, and see what differences they will make for the final performance.
> > > > > We will leave the results and analysis in a further version.

---

### Official Review · Reviewer_BP2L · 2022-10-25

**Confidence:** 4
**Correctness:** 3
**Technical Novelty And Significance:** 3
**Empirical Novelty And Significance:** 3
**Recommendation:** 6

**Clarity, Quality, Novelty And Reproducibility:**

I find the composable task codes novel and the paper is well written. I believe results can't be reproducible without proper details on hyper parameters.

**Strength And Weaknesses:**

**Strengths** The paper studies a very important and interesting problem -- learning composable task codes for generalization to unseen tasks. I find the formulation of the problem using VQ-VAE natural and inference methods convincing. It is interesting to see that empirical results, especially zero-shot learning, improves compared to self-training or T0 models.

**Weaknesses** There are a few points that require clarification and improvement.
1. Does sharing a code necessarily mean that two tasks are similar in some sense? I found no evidence as to how important a code is for a task; hence, I can't say if a code is merely noise that gets ignored by the model eventually. For example, if you increase the number of codes, *L*, the model might be given more codes but not all of them would be used. There is not loss that encourages the LLM to utilize all codes as well.

2. What is the effective number of codes that discrete compositional task code, *z*, has? Are all codes in z unique or are they frequently reused?

3. Comparison to previous work needs clarification.
- Are baselines multi-task learners or are they initialized with some pretrained embedding? Like, prompts can be initialized or they can be trained in mutli-task fashion.
- What are the architectural details of other models, including model size, are they comparable?
- What is the performance of model tuning if you use all 32 examples for training? I think it is a bit unfair to use all examples for CTR while using only half of them for others.
- You mention that other methods update extra parameters as a weakness but this is not true. Prompt tuning only updates an additional prompt while the LLM is fixed. On the other hand, CTR updates a codebook and LLM is also fine-tuned. I think this is less scalable overall compared to parameter efficient updates.

4. How do you choose hyperparameters? Like, "CTR length", "codebook size", or *N=60*. I don't see a clear pattern on final performance and no training/dev results are given.

5. Could you clarify:
- How do you combine with manual prompts? Are they appended as additional soft-prompt vectors?
- What is the performance if you haven't updated the LLM? This would be a fairer comparison to parameter efficient models.
- What is the performance w.r.t. increasing data size for few-shot learning?
- Why does the model perform worse on co-reference?

**Summary Of The Paper:**

This paper studies learning composable task codes for soft-prompting of language models. The authors propose an approach similar to VQ-VAE where each task is associated with a set of discrete codes and embedding of these codes from a codebook lookup table is stacked as a soft-prompt for a language model. First, a 2D embedding for each training task is learned via another lookup table. Next, each row of this embedding matrix is used to search for the nearest code embedding in codebook lookup table. The model is trained with typical language model loss in addition to commitment and embedding losses for learning codebook embeddings. By sharing codes across tasks, this approach presents a natural task compositionality. The authors propose code ensembling for zero-shot learning and bitwise searching for few-shot learning. On a set of benchmark tasks, the model compares favorably to previous models; achieving on-par or better on average. The authors also experiment with interpretability -- similar tasks share codes -- and controllability -- bitwise perturbation exhibits different behaviors.

**Summary Of The Review:**

I think composable task codes, VQ-VAE style training for LLM fine-tuning, and generalization to novel tasks during inference are strong points of the paper. But, there are still pieces that are unclear and needs improvement.

---

> ### Author Response · Authors · 2022-11-19
> **Response to Reviewer BP2L**
>
> We thank the reviewer for the insightful feedback. We would like to clarify a few points as follows.
> 1. Response to #1:
>     * Yes, similar tasks share the same code. After training the compositional task code, we performed a qualitative analysis of the training tasks and found that similar training tasks share similar codes. So we inherited the conclusion to the test code selection.
>     * We have added the performance of different selections of codes in Appendix A.7. Results show that the zero-shot performance is quite sensitive to the selection of codes (even up to 20 points on certain tasks), proving that the code is definitely not noise.
> 2. Response to #2:
>     * In our experiments, we use a codebook size of 128. Intuitively, a small codebook size will not provide sufficient capacity for task representations, while a large codebook size will cause inefficiency. We determine the codebook size according to practical considerations. In addition, our ablation study on codebook size confirms the empirical selection.
>     * Besides, we have added statistics in Appendix in terms of the usage frequency of different codes. Results show that most of the codes are used.
> 3. Response to #3:
>     * Baselines are also multi-task learners. So they are fair to directly be compared.
>     * Yes, all the baselines have a comparable model size as ours, which is 770M.
>     * Good point! In our paper, the fairness lies in that all methods have only 32 labeled examples in total. The model tuning needs to perform model selection, so it has to split a dev set to select the best checkpoint, leaving fewer labeled examples for training. Our CTR does not finetune the test data, thus no need for model selection. It can use all 32 labeled examples to select code.
>     * There could be a misunderstanding. Our setting focuses on cross-task generalization. So there are two phases---phase 1 improves cross-task generalization without seeing the test task, whereas phase 2 works on the test task. For a fair comparison, we ensure all baselines are multi-task trained in phase 1, such that their cross-task generalization is all boosted through multi-task learning. (1) For our CTR, in phase 1, it first trains codebook task representations and then trains the LLM along with the codebook task representations. In Phase 2, it searches code and evaluates test tasks without any parameter updates. (2) For promptuning, in phase 1, it first multi-task trains an LLM, obtaining T0. Phase 2 uses a few test-task examples to optimize the free parameters. We do accept that the original expressions could be misleading, and have adjusted the expression.
> 4. Response to #4: We have added detailed descriptions in our paper. For hyper-parameters shared by T0, we directly followed the same training recipe as T0. For unique hyper-parameters such as CTR length and codebook size, we determine values according to practical considerations (i.e., empirical values).
> 5. Response to #5:
>     * We have added details of how to combine with manual prompts in Section 4.3.1. Specifically, for experiments without manual prompts, the input texts are a direct concatenation of multiple text fields, with CTR appended in front of it. For experiments with manual prompts, the input texts are constructed by leveraging natural language prompts, with CTR placed in front.
>     * Same as Response #3(c).
>     * Good point. We are conducting experiments and will add detailed results in a further version.
>     * Good question. Results show that, on co-reference tasks, CTR performs worse under the zero-label setting but better under the few-shot setting. The difference is that the zero-label setting uses pseudo-labeled data to select test codes while the few-shot setting uses real-labeled data to select codes, both sharing the same checkpoints. The reason for decreased performance lies in that the pseudo data of the co-reference task were of low quality and they did not select effective task codes.

---

> > ### Comment · Reviewer_BP2L · 2022-11-28
> > **Thank you for the response.**
> >
> > 1- I think it makes sense that the model is sensitive to completely different codes, even if there is small overlap, but this doesn't necessarily mean that there are no redundancies in codes that are chosen by your method. It would be more helpful if you use your method to select codes for a test task and do an ablation by removing each code.
> >
> > 2- How do you choose codebook size and CTR length? You said "practical considerations" and in your response #4 you mentioned "empirical performance". It seems the ablation in Table-11 and main results in Table-1 match, suggesting that you chose these hyperparameters using the test examples.
> >
> > This also means that for zero-label setting, you are using some labeled data to search for the hyperparameters.
> >
> > This is also related to my fairness question. You need to select hyperparameters that are specific to your model (not different from model tuning). I think it is necessary to use 16 examples to choose codes and remaining 16 examples to choose hyperparameters to make a fair comparison.

---

> > > ### Author Response · Authors · 2022-12-01
> > > **Response to Reviewer BP2L**
> > >
> > > We thank the reviewer for the constructive feedback.
> > >
> > > Response to comment #1:  Currently, the main focal point of our work lies in proving its effectiveness, with coding efficiency unconsidered. That's to say, the method is considered effective as long as it learns the task feature at least in one code.
> > > Indeed, there could be redundancies in codes. We do agree that the study of coding efficiency is a valuable follow-up research question. We humbly take the advice to conduct further ablation of removing each code, such that one can understand the learned codes better.  We will leave it for future work.
> > >
> > > Response to comment #2: please see the [response to Reviewer zu2t](https://openreview.net/forum?id=6axIMJA7ME3&noteId=HIkbxNSZ0CL).
> > > Briefly, for a fair comparison, we follow exactly the same method of determining hyper-parameters as our baseline T0. We experimented with a set of hyper-parameters, chose the best-performing one, and then fixed them during all experiments. We agree that such a method could be somewhat problematic, since it is exposed to the risk of overestimation, to a certain degree, for both baselines as well as ours.
> > >
> > > We agree that using 16 examples (i.e., few-shot dev set) to choose hyper-parameters is a potential method. We will leave the work in the future version. Thank you!

---

### Public Comment · ~Anirudh_Goyal1 · 2022-11-15
**Good execution**

Hello,

I like the idea  as well as the execution of the idea in this paper. This is good work! :)

In our work, we have explored "learning a discrete, compositional/factorial codebook" for various different problems and shared the similar motivation as here i.e., by training on compositions of many codes, it may be possible to generalize to novel combinations (that said, we have not explored this on the dataset/problem which this paper is trying to do).

- [Discrete-Valued Neural Communication (NeurIPS'21)](https://arxiv.org/abs/2107.02367): Learning a factorial codebook helps different components in multi-component architecture (like slots in slot-based architecture, positions in transformers, and nodes in graph based methods) generalize systematically.

- [Discrete Key Value Bottleneck (arXiv)](https://arxiv.org/abs/2207.11240): Basic idea is to have a multi-head factorial codebook of learnable (key, value) pairs, where the information about the input is bottleneck via the learnable factorial codebook (key-value pairs). Here, the keys are "optimised" with respect to the encoder, and the "values" are optimised with respect to downstream decoder. During adaptation, we only adapt a subset of the values to achieve rapid adaptation, whereas everything else is kept fixed (here, the adaptation is task-agnostic).

- [Discrete Factorial Representations as an
Abstraction for Goal Conditioned RL (NeurIPS'22) ](https://arxiv.org/abs/2211.00247): [There's no way authors could have been aware of this work as it came on arXiv after the deadline) Recently, we also explored the use of such a compositional bottleneck for visual goal-based reasoning, where we tried to focus on "goal-specification" using the learned compositional codebook.


Thank you for your time in reading my message.

---

> ### Author Response · Authors · 2022-11-19
> **Thank you for your interest in our work**
>
> Hi! Thank you for your interest in our work and providing valuable references.
>
> We guess the theoretical analysis in the DVNC paper can also explain why a discrete, compositional codebook can improve the cross-task generalization ability of large language models. We will continue working in this direction.

---

### Decision · Program_Chairs · 2023-01-20

**Decision:**

Accept: poster

**Justification For Why Not Higher Score:**

The evaluation results are on the optimistic side since the setup is not truly zero shot as it involved labeled data. (See summary above)

**Justification For Why Not Lower Score:**

The proposed approach is interesting and well-received by the reviewers.

**Metareview: Summary, Strengths And Weaknesses:**

This paper proposes a prompt-free method of inducing a language model to perform a new task. Specifically, a set of code vectors are learned on a collection of training tasks. The paper then proposes a label-free and few-shot method for selecting which codes to use on an unseen task. Evaluation is primarily done in the same setting as T0 with improved performance compared to other prompt-learning-style methods, along with analysis of the learned codewords and their usage. While reviewers generally agreed that the method was interesting and novel, one reviewer pointed out an important issue with the paper - the proposed method is not truly zero-shot because a wide variety of different variants of the algorithm (including different hyperparameters and architectural designs) are evaluated on the held-out tasks and performance is reported for the best variant. This is not zero-shot learning because it involves using labeled data to infer the best system design. Using labeled data for system design in a zero-shot setting is equivalent to training on the test set (see e.g. "True Few-Shot Learning with Language Models") and could be very misleading to the readers. We strongly recommend the authors to revise the terminology in the final version of the manuscript to truthfully reflect the actual evaluation setup in order to avoid unnecessary confusion and misunderstanding.

**Note From Pc:**

if the above contains the word "oral" or "spotlight" please see: "oral" presentation means -> notable-top-5% and "spotlight" means -> notable-top-25%. As stated in our emails, we are disassociating presentation type from AC recommendations